# Catastrophic Antiphospholipid Syndrome

**DOI:** 10.3390/ijms25010668

**Published:** 2024-01-04

**Authors:** Victoria Bitsadze, Fidan Yakubova, Jamilya Khizroeva, Arina Lazarchuk, Polina Salnikova, Alexander Vorobev, Maria Tretyakova, Natalia Degtyareva, Kristina Grigoreva, Nilufar Gashimova, Margaret Kvaratskheliia, Nataliya Makatsariya, Ekaterina Kudryavtseva, Anna Tomlenova, Jean-Christophe Gris, Ismail Elalamy, Cihan Ay, Alexander Makatsariya

**Affiliations:** 1Department of Obstetrics, Gynecology and Perinatal Medicine, N. F. Filatov Clinical Institute of Children’s Health, I. M. Sechenov First Moscow State Medical University (Sechenov University), Trubetskaya Str. 8-2, 119991 Moscow, Russia; fidan.yagubova.2021@gmail.com (F.Y.); totu1@yandex.ru (J.K.); arina.lazarchuk@mail.ru (A.L.); salnikovapolina@bk.ru (P.S.); alvorobev@gmail.com (A.V.); tretyakova777@yandex.ru (M.T.); soba4ka-10@yandex.ru (N.D.); grigkristik96@gmail.com (K.G.); nelya.94@yandex.ru (N.G.); margaret.kv@mail.ru (M.K.); makatsariya@gmail.com (N.M.); katena.kudryavtseva.2000@mail.ru (E.K.); anna001anya@gmail.com (A.T.); jean.christophe.gris@chu-nimes.fr (J.-C.G.); ismail.elalamy@aphp.fr (I.E.); cihan.ay@meduniwien.ac.at (C.A.); 2Faculty of Pharmaceutical and Biological Sciences, Montpellier University, 34093 Montpellier, France; 3Department Hematology and Thrombosis Center, Medicine Sorbonne University, 75012 Paris, France; 4Hospital Tenon, 4 Rue de la Chine, 75020 Paris, France; 5Department of Medicine I, Clinical Division of Hematology and Hemostaseology, Medical University of Vienna, 1080 Vienna, Austria

**Keywords:** antiphospholipid antibodies, antiphospholipid syndrome, complications of pregnancy, catastrophic antiphospholipid syndrome

## Abstract

Unlike classic APS, CAPS causes multiple microthrombosis due to an increased inflammatory response, known as a “thrombotic storm”. CAPS typically develops after infection, trauma, or surgery and begins with the following symptoms: fever, thrombocytopenia, muscle weakness, visual and cognitive disturbances, abdominal pain, renal failure, and disseminated intravascular coagulation. Although the presence of antiphospholipid antibodies in the blood is one of the diagnostic criteria, the level of these antibodies can fluctuate significantly, which complicates the diagnostic process and can lead to erroneous interpretation of rapidly developing symptoms. Triple therapy is often used to treat CAPS, which includes the use of anticoagulants, plasmapheresis, and high doses of glucocorticosteroids and, in some cases, additional intravenous immunoglobulins. The use of LMWH is recommended as the drug of choice due to its anti-inflammatory and anticoagulant properties. CAPS is a multifactorial disease that requires not only an interdisciplinary approach but also highly qualified medical care, adequate and timely diagnosis, and appropriate prevention in the context of relapse or occurrence of the disease. Improved new clinical protocols and education of medical personnel regarding CAPS can significantly improve the therapeutic approach and reduce mortality rates.

## 1. Introduction

More than 40 years after discovery, APS remains one of the most mysterious syndromes in modern medicine. APS is a heterogeneous systemic syndrome and at the same time an autoimmune disease, acquired immune thrombophilia, and thromboinflammatory disease.

Clinical manifestations are caused by antibody-mediated activation of key target cells and modulation of several major biological systems through the interaction of these antibodies with various cofactors and special cell surface receptors. Such interactions lead to the activation of complement on cell surfaces, activation of neutrophils and monocytes, release of antiangiogenic factors, reactive oxygen species (ROS), TNF-a, activation of blood coagulation, inflammation, and NETosis (neutrophil extracellular trap formation). Because of the variable clinical manifestations of APS, it is often called “chameleon syndrome”. However, not all clinical manifestations are included in the criteria for APS, nor are antiphospholipid antibodies, which are divided into criteria (lupus anticoagulant (LA), anticardiolipin (ACL), anti-β2-glycoprotein I (anti-β2GPI)) and non-criteria (antiannexin A2, antivimentin/cardiolipin complex, antiannexin A5, antiphosphatidylethanolamine, antiphosphatidylinositol, etc.). Therefore, the classification criteria for APS are revised from time to time.

The so-called Sydney or revised Sapporo criteria for APS are very well known and include clinical criteria (venous and/or arterial thrombosis, and/or microcirculatory thrombosis, and/or morbidity in pregnant women) and laboratory criteria (LA, and/or aCL, and/or anti-β2GPI antibodies). APS is confirmed if at least one clinical and one laboratory criteria are present. In 2023, new classification criteria for APS were developed and proposed by the American College of Rheumatology (ACR) and the European League Against Rheumatism (EULAR). The supplemented clinical criteria are divided into six domains and laboratory criteria into two domains. The proposed new APS criteria are highly specific (99% versus 86% of the current Sapporo criteria), risk stratified, and hierarchically clustered [1].

Due to the heterogeneity of clinical manifestations and pathogenetic mechanisms, APS is conventionally divided into “pure” obstetric (o-APS) and thrombotic antiphospholipid syndrome (t-APS). O-APS is characterized by recurrent fetal loss, intrauterine death, neonatal death, placental insufficiency, preterm birth, placental abruption, severe preeclampsia, and HELLP syndrome. Thrombosis is less typical for o-APS, although o-APS and t-APS can be combined. Currently, pediatric and even neonatal APS are also distinguished. Children may also develop an extreme form of APS, catastrophic antiphospholipid syndrome (CAPS), as the first manifestation of the syndrome.

Catastrophic antiphospholipid syndrome (CAPS) is a rare and life-threatening condition, characterized by thromboses with the development of multiple organ failure [2]. The classification criteria for CAPS are: damage to three or more organs, rapid development of clinical manifestations, histopathological patterns of small vessel occlusion and the circulating APLA (Table 1) [3,4]. The disease was first described by S. Greisman in 1991. However, this pathology is also called Asherson syndrome in honor of the doctor who first introduced the term “CAPS” in 1992 [5]. In the 1980s, while working at the Hammersmith hospital, Ronald Asherson began collecting and describing rare cases of CAPS. Together with Ricardo Cervera, they initiated the creation of a worldwide database. The result of their work was the CAPS international register, where everyone can add clinical cases [6]. Although CAPS develops in less than 1% of patients with APS, it is a life-threatening condition that requires early diagnosis and immediate treatment [7].

During pregnancy, an additional organ, placenta, appears, therefore placental thrombosis should also be considered in the diagnosis of CAPS.

The data sources for primary articles were the following databases: PubMed, Web of Science, the Cochrane Database, Wiley Online Library, ScienceDirect, Elibrary, Medline, ResearchGate, and Dissertation Abstracts International. The search strategy used a combination of free text search, Medical Exploded Subject Headings (MESH), and all synonyms of the following terms: “catastrophic antiphospholipid syndrome” and “CAPS” with “pathophysiology”, “treatment”, “epidemiology”, “diagnosis”, and equivalent terms in other languages.

## 2. Risk Factors and Pathogenesis

Extreme care and vigilance are required to ensure that the disease is not missed if a patient has risk factors for CAPS. The main risk factors include infection, surgery or trauma, anticoagulant withdrawal, pregnancy and postpartum period, combined oral contraceptive intake, vaccination, ovulation induction in ART, and many other factors [8]. Genetic thrombophilia in pregnant women with APS is also a significant risk factor for the development of CAPS, especially in the 3rd trimester and in the postpartum period [9]. The European League Against Rheumatism (EULAR) recommendations defined a high-risk thrombotic profile as the presence of LA or double/triple APLA positivity or the presence of persistently high APLA titers [10]. However according to the re-evaluated risk profile of the obstetric APS, low APLA titers confer risk for pregnancy morbidity in addition to the widely accepted risk associated with medium/high APLA titers; persistent double positivity for LA and low titers of anti-β2GPI IgG display a higher risk than low aCL and anti-β2GPI IgG, either alone or associated [11].

APLA affects many elements of the hemostatic system, and the main pathogenetic mechanisms can be divided into four groups:(1)cellular activation (endothelial, immune cells, platelets),(2)inhibition of anticoagulant potential,(3)inhibition of fibrinolysis,(4)activation of complement system [12,13] (Figure 1).

As a result of the presence of APLA, stimulation of endothelial and immune cells, as well as platelets, occurs. The interaction of APLA with anionic determinants (phospholipids of cell surfaces of endothelial cells, throphoblasts, etc.; polyphosphates of platelets or nucleic acids, etc.) requires the presence of β2GPI. β2GPI is the principal cofactor of APLA and exists in two different conformations: circular and open (activated) [14]. In the circular conformation, the epitope is protected from the plasma, but after interaction with anionic surfaces the epitope becomes exposed. The formation of antibodies with these epitopes is crucial in the pathogenesis of APS and thrombogenesis [15]. The distribution of β2GPI has distinct features. It is not present on resting endothelium but can be found on the vascular endothelium only after an inflammatory stimulus. However, it is present on the endothelium of uterine vessels and trophoblast, as well as in placental tissues at the implantation site in a normal pregnancy [16,17,18].

As a result of p38 mitogen-activated kinase (p38-MAPK) and the activation of the transcription factor NF-κB, a proinflammatory and prothrombotic response develops. It manifests by an expression of adhesion molecules (E-selectin, ICAM-1, VCAM-1) and the production of cytokines (TNF-a, IL-1b, IL-6). Cytokines activate lymphocyte adhesion that contributes to further progression of inflammation. APLA also leads to decreased activity of endothelial nitric oxide synthase (eNOS). Interacting with the apolipoprotein E2 receptor (ApoER2), APLA results in decreased nitric oxide production. NO deficiency leads to impaired vasodilation and promotes platelet adhesion [19]. Circulating anti-β2GPI also leads to activation and increased expression of tissue factor (TF) on the surface of endothelial cells and monocytes. TF, composed of the apolipoprotein C-III and a phospholipid complex, is a cell surface receptor and cofactor for coagulation factor VII (FV VII) [20].

APLA’s persistence also leads to the inhibition of endogenous anticoagulants such as protein C (PC), protein S, prothrombin, and annexin [21,22]. Activation of PC occurs due to the binding of thrombomodulin and thrombin on the surface of endothelial cells and platelets. APLA inhibits protein C by:(1)inhibition of PC complex assembly,(2)reducing the activation of PC through the thrombomodulin–thrombin complex,(3)direct suppression of PC activity,(4)decreased clearance of PC.

Annexin V, which has a high affinity for phospholipids (PLs), is a natural anticoagulant and a marker of apoptosis. The phospholipid membranes of apoptotic cells are bound to annexin V, that prevents the formation of autoantibodies and a possible excessive procoagulant response. APLA can destroy the “annexin shield” when competing with PLs such as phosphatidylserine, that provokes the development of an autoimmune response [23].

APLA can also influence the extrinsic and intrinsic fibrinolysis pathways (Figure 2). Excessive activity of the coagulation system is regulated by complex interactions of activators and inhibitors, cofactors, and receptors. Plasminogen is converted to plasmin through the action of tissue (tPA) and urokinase (uPA) plasminogen activators. Plasminogen activator inhibitor-1 (PAI-1) inhibits the activity of both tPA and uPA. APLA leads to endothelial activation, resulting in the release of increased amounts of tPA and PAI-1. By interacting with annexin A2 (AnnA2), APLA prevents tPA from binding to its endothelial receptor. APLA potentially interferes with the normal function of each of the fibrinolytic proteins, contributing to the hypofunction of the fibrinolytic system [24,25].

The complement system is also involved in the pathogenesis mechanism. In patients with the acute phase of the disease, the levels of complement C3 and C4 are decreased, but complement activation products, for example, C5b-9, are consequently increased [26]. Complement activation mainly occurs by the classical pathway. As a result of the action of C3 convertase, the C3 component is split into C3a and C3b. On the surface of platelets, C3a binds to its receptor. As a result, platelet activation, adhesion, and aggregation occur. C3b promotes phagocytosis and is involved in the assembly of C5 convertase, which cleaves C5 into C5a and C5b. C5a stimulates the expression of tissue factor (monocytes, neutrophils, endothelial cells) and PAI-1 (mast cells, basophils). C5b is involved in the assembly of the membrane attack complex (C5b-9) and its deposition on the cell surface, which leads to the cell damage. Chaturvedi et al. [27] demonstrated that patients with CAPS have a higher incidence of rare germline variants in complement regulatory genes (60%) as well as a higher frequency of mutations in these genes.

However, there is still debate about why CAPS occurs even in the presence of long-term APLA. The “two-hit” theory suggests that there must be an additional biological triggering or precipitating factors leading to endothelial activation, massive cytokine release, microvasculopathy, and thrombosis (Figure 3).

Molecular mimicry is one of the CAPS development mechanisms. According to research results, a number of microorganisms contain special sequences in their genetic material that are similar to the sequences in the binding site of β2GPI with phospholipids, resulting in cross-reactivity with the own structures [28,29].

There is growing evidence suggesting a key role for systemic inflammatory response syndrome (SIRS). SIRS occur not only with an infectious lesion but also with a number of autoimmune diseases. One of the SIRS manifestations is respiratory distress syndrome, that is very common in patients with CAPS (up to 25%). It is proved by the presence of common mediator cascades characteristic of both CAPS and sepsis [30]. Pregnancy is associated with changes in hemostasis, leading to hypercoagulability, and at the same time is accompanied by an increase in proinflammatory molecular markers and the development of compensated systemic inflammatory response [31]. In the presence of APLA, the altered cross-talk between inflammation and coagulation can form a second hit, leading to the development of decompensated SIRS and the thrombotic storm in CAPS.

According to studies, the most common histopathological characteristic observed in CAPS is intravascular thrombosis of small vessels. Renal biopsy specimens often present acute thrombotic microangiopathy (TMA). This condition is characterized by the presence of blood clots in the arterioles, fibrous intimal hyperplasia, and focal cortical atrophy [32]. Immune complex deposits are rarely observed in patients with CAPS [33,34].

## 3. Clinical Manifestations

According to a descriptive analysis of 500 patients included in the registry, CAPS was more common in women (69%). Forty percent of patients had associated autoimmune diseases such as SLE. The majority of CAPS episodes were triggered by various factors, with infection being the most frequent one (49%). Almost all organs and systems can be involved in CAPS manifestation. However, the skin, lungs, kidneys, central nervous system, heart, and gastrointestinal organs are the most affected [35].

Dupré et al. [36], in a cohort study, assessed cutaneous manifestations in 65 patients with CAPS. The most common skin lesions observed were livedo racemosa (*n* = 29, 45%), the second most common were necrotic lesions and ulcerations (*n* = 27, 42%). Spontaneous subungual hemorrhages (*n* = 19, 29%), swelling of the distal phalanges (*n* = 15, 23%), and purpura (*n* = 9, 14%) were also observed. During the acute stage of CAPS, 16 histological studies were performed. In 94% of cases, biopsy results showed thrombosis of the skin capillaries. Basically, all skin manifestations of CAPS resolved without complications.

Sanchez et al. [37] described a case of CAPS involving the skin, eyes, and brain. A 17-year-old patient developed fatigue, loss of appetite, ulcerative skin lesions of the lower extremities, and livedo reticularis. The results of an ophthalmological examination revealed ischemic damage to the retina in the temporal quadrant, as well as perivascular hemorrhages of the ocular fundus. MRI of the brain showed several lacunar white matter infarcts and small cortical/subcortical and subarachnoid hemorrhages. After 2 months of intensive therapy, the patient’s condition improved.

Mardani et al. [38] described a case of CAPS in a woman with psychosis and long-standing hypertension and a history of recurrent miscarriage. The patient was admitted to the hospital with complaints of confusion, asterixis, uncontrolled blood pressure, nausea, and vomiting. Laboratory tests results suggested acute kidney damage, and hemodialysis was started. A kidney biopsy revealed glomerular sclerosis, fibrinoid necrosis, thrombosis with partial or complete closure of the lumen of the interlobular arteries, and edema of the vessel walls. The results of immunofluorescence analysis revealed the activity of the C3 complement component. Among APLAs, a positive IgG aCL was noted. A diagnosis of APS nephropathy was established based on morphological, clinical, and laboratory data. aCL were also found after 14 weeks, confirming the diagnosis of CAPS.

Oredegbe et al. [39] presented a difficult-to-diagnose case of CAPS in a 51-year-old woman with a fulminant type of disease. The patient was initially hospitalized with abdominal pain, and a full examination revealed hemorrhage in the adrenal glands and multiple thromboses affecting many organ systems. This clinical case is considered particularly difficult to diagnose, since bilateral adrenal hemorrhage was the first manifestation of CAPS in the patient. To ensure successful treatment of this condition, it is crucial to consider the possibility of CAPS in patients presenting with multiple organ failure, multiple thromboses, and bleeding.

The retrospective study included 23 patients with CAPS and the posterior segment of the eye lesions. Ninety-one percent of patients were women with an average age at diagnosis of 28 years. Ophthalmic manifestations in CAPS were usually bilateral (*n* = 19, 83%). The most commonly reported manifestations were occlusive retinopathy (*n* = 17, 74%) and chorioidopathy (*n* = 11, 48%). Retinal vasculitis was reported in only one case (*n* = 1, 4%). After a long-term follow-up of 14 months, almost half of the patients (*n* = 11, 48%) experienced permanent vision loss [40].

Most often, CAPS develops in patients with systemic lupus erythematosus (SLE) (up to 30%) [35], described figuratively as a “tsunami” in the ocean. Dorji et al. [41] presented three cases of fulminant CAPS in patients with SLE with different clinical manifestations in all affected women. A 22-year-old woman with SLE was admitted to the hospital with manifestations of anasarca and thought disorders. Laboratory tests revealed anemia, thrombocytopenia, and positive testing for ANA and LA. Pulmonary embolism and infarction of the left middle cerebral artery developed. Despite adequate therapy, the patient experienced a refractory course of cytokine storm and neutropenic sepsis and died. In the second case, a 23-year-old woman was admitted to the hospital with oral aphthae, polyarthritis of the small joints of the hands, Raynaud’s syndrome, intermittent fever with headache, and a history of arterial thrombosis. The patient’s CAPS occurred with generalized tonic–clonic seizures and papilledema. MRI of the brain revealed thrombosis of the superior sagittal sinus. After adequate therapy, the acute condition resolved. A 47-year-old woman with SLE and a history of class IV lupus nephritis was admitted to the hospital with paraparesis and absence of pulses in the femoral, popliteal, and anterior tibial arteries. After CT angiography, thrombosis of the abdominal aorta and left popliteal artery was revealed. Laboratory test results showed high titers of anti-β2GPI IgM. The patient was treated with methylprednisolone, anticoagulants, and broad-spectrum antibiotics, but after the first cycle of plasmapheresis she experienced a sudden deterioration in hemodynamics, thrombosis of the anterior spinal artery, and cardiac arrest, as a result of which the patient died. Due to the development of CAPS, all patients exhibit a polymorphism of clinical manifestations, which significantly complicates prompt diagnosis.

Neonatal antiphospholipid syndrome (APS) also exhibits unique characteristics. Catastrophic antiphospholipid syndrome (CAPS) is one of the most severe and frequent initial thrombotic events in APS, with a poor prognosis. It remains uncertain whether there are differences in outcomes between de novo and transmitted neonatal APLA. There are few descriptions of neonatal catastrophic antiphospholipid syndrome (CAPS) in the literature [42,43]. In clinical practice, the condition may be misdiagnosed. APLA testing in such cases may be useful to rule out CAPS.

However, CAPS does not always develop in patients with diagnosed SLE. Carvalho et al. [44] described the case of a 13-year-old girl who was admitted to the hospital with acute peritonitis secondary to ischemic perforation of the sigmoid colon. Angiography results revealed thrombosis of the superior mesenteric artery. LA and ANA were detected in the blood serum. The diagnosis of CAPS was based on fulminant damage to four organs: intestines, lungs, kidneys, and spleen. After successful therapy, she was discharged from the hospital, but 2 months later the patient was diagnosed with SLE due to ongoing circulation of LA.

Nawata et al. [45] presented a unique case of CAPS in a 14-year-old girl with Epstein–Barr virus (EBV)-associated hemophagocytic syndrome. The patient developed renal, intestinal, and pulmonary infarction, as well as thrombocytopenia and hemolytic anemia within 1 week. She was initially diagnosed with TMA and given appropriate treatment. However, despite the therapy, the patient’s platelet count decreased and her condition worsened, leading to death. An autopsy revealed multiple renal infarctions on both sides, as well as jejunum, ileum, and pulmonary parenchyma infarctions. Massive hemophagocytic lymphohistiocytosis was observed in the splenic pulp, lymph nodes, and bone marrow. EBV was detected in the bone marrow, and the presence of antibodies indicated previous infection. Testing for APLA was also positive.

Intestinal involvement in CAPS may be associated with an even greater increase in mortality. Christiansen et al. [46] described the case of a young man admitted to the hospital with abdominal pain and vomiting. CT results revealed pneumoperitoneum, and diagnostic laparoscopy disclosed a pattern of necrosis of the small intestine, which indicated thrombosis of small vessels. No colon damage was observed. After adequate therapy, the patient was transferred to a rehabilitation center.

Patients with genetic thrombophilia may develop more severe forms of CAPS. McRae et al. [47] described a case of CAPS in a 25-year-old female patient with the prothrombin gene mutation G20210A and heterozygous Leiden. She took combined oral contraceptives for 6 months. The patient was admitted to the hospital with the neck and right shoulder pain worsening with movement. The next day she developed shortness of breath and chest pain. CT results revealed pulmonary embolism and pericardial effusion with enlarged regional lymph nodes. A repeat ultrasound in the intensive care unit revealed thrombi in the right subclavian/axillary vein and left axillary vein. Despite the therapy, the patient’s condition worsened. On the third day, cerebral edema occurred with two areas of subacute infarction and necrotic changes in both upper extremities, which required amputation. Positive LA titers were detected in the blood. After excluding the diagnoses of thrombotic thrombocytopenic purpura (TTP) and heparin-induced thrombocytopenia (HIT), genetic testing was performed. The results of histopathological examination of the skin revealed necrosis of the epidermis and thrombosis of small vessels. The likelihood of thrombosis in women taking combined oral contraceptive pills (COCPs) increases 3–5 times.

The impact of CAPS on pregnancy has been the subject of growing research interest. CAPS most often develops in the last trimester or in the postpartum period [48]. Most patients had previously experienced signs of APS, such as fetal loss or thromboembolism. Due to the similarities in the clinical presentation of CAPS and obstetric complications such as HELLP syndrome (hemolysis, elevated liver enzymes, low platelet counts) or acute fatty liver of pregnancy (AFLP), differential diagnosis of these conditions can be difficult. As a result, the risk of adverse maternal and perinatal outcomes increases [49]. According to Hoayek et al. [50], when CAPS occurs during pregnancy or in the postpartum period, an earlier age of onset of the disease is observed. APLAs have a negative effect on the formation and functioning of the placenta due to disruption of the remodeling of the spiral arteries, inflammation of the decidua, reduction in the number of syncytial nodules in the placental villi, and syncytial capillary membranes. Placental dysfunction is the cause of the development of obstetric complications such as pre-eclampsia, HELLP syndrome, and placental abruption. The appearance of these complications in patients with APS is always suspicious for CAPS. As CAPS is a systemic pathology, it causes damage not only to the maternal body and placenta but also to the fetus. According to a study, there is an association between SIRS in the fetus and fetal thrombophilia. In cases of chronic infection and the presence of systemic inflammatory response syndrome, the fetus may develop SIRS [51]. According to studies, in the pathogenesis of complicated pregnancy and reproductive losses, a key role is played not only by maternal thrombophilia but also by fetal thrombophilia [52,53].

Gómez-Puerta et al. [54] described 15 patients with CAPS with a history of obstetric complications. In half of the cases, CAPS occurred during pregnancy, in six (43%) in the postpartum period, and in one patient after curettage as a result of intrauterine fetal death. The survey revealed several specific features. In eight (53%) patients, a diagnosis of HELLP syndrome was established, and in four (27%) placental infarction was observed. Pelvic vein thrombosis and myometrial TMA occurred in one patient. Mortality rates were high among both mothers (46%) and infants (54%).

One of the infrequent manifestations of CAPS is bone marrow necrosis. Typically, this disease develops in patients with neoplastic processes, lymphomas, sickle cell anemia, or as a result of radiation or chemotherapy. However, CAPS should also be added to this list. Many authors have described clinical cases of bone marrow lesions in young women [55,56]. Sinha et al. [57] described a case of extensive and fatal bone marrow necrosis involving the liver, lungs, and central nervous system in a 22-year-old woman during pregnancy with very high APLA titers (Table 2).

## 4. Diagnostic Difficulties

Due to the variety of CAPS clinical manifestations, it can be difficult to diagnose (Table 3). Huang et al. [58] suggested using a history of arterial hypertension, anemia (OR 116.231, 95% CI 10.512–1285.142), elevated LDH levels (OR 59.743, 95% CI 7.439–479.815), and proteinuria (OR 11.265, 95% CI 2.118–59.930) in laboratory test results as an early predictor model in patients with an established diagnosis of APS.

Ferritin is an important signaling molecule and a direct mediator of the immune system. There is a close association between hyperferritinemia and such severe conditions as CAPS, macrophage activation syndrome, and septic shock. In a study by Rosário et al. [59], 71% of patients with catastrophic APS had high serum ferritin levels (more than 1000 ng/mL).

If a patient has positive APLA titers, the diagnosis of APS cannot always be established. According to modern data, APLAs are present in the plasma of 1–5.6% of healthy people, and the antibody titer increases with age [60]. The triggering mechanisms and causes of the presence of APLA have not yet been studied, as well as the full range of possible clinical manifestations. The presence of APLA may be a sign of inflammation or cancer, but further development of thrombosis does not always occur [61,62].

Sepsis is an extremely dangerous condition that develops in response to an infection.

The presence of two or more of the following symptoms of SIRS is necessary to confirm sepsis.

(1)Body temperature more than 38 °C or less than 36 °C;(2)Pulse rate more than 90/min;(3)Respiratory rate more than 20/min or PaCO_2_ less than 32 mmHg (4.3 kPa);(4)Laboratory tests: white blood cell count more than 12,000/mL or less than 4000/mL or more than 10% immature cells [63].

Distinguishing sepsis from CAPS is not always easy [64,65]. Tucker et al. [66] presented a case report of a patient diagnosed with probable CAPS who developed inflammatory bowel disease with bloody diarrhea and abdominal pain. The patient’s medical history indicated a history of spontaneous abortions. Further examination showed positive titers of LA and portal vein thrombosis, which led to intestinal ischemia. However, after starting a course of therapy with steroids and anticoagulants, the patient developed bacteremia and candidiasis, which required discontinuation of steroids. Despite the use of antibacterial drugs, the patient’s condition worsened, she developed multiple central venous thromboses, as well as occlusion of the right ovarian vein, which subsequently led to death. When sepsis is combined with DIC syndrome, patients may develop thrombocytopenia, bleeding, and thrombosis of small vessels, which ultimately leads to the development of multiple organ failure [67].

Heparin-induced thrombocytopenia (HIT) is an immune form of thrombocytopenia caused by unfractionated heparin (UFH) or, rarely, low molecular weight heparin (LMWH) use. In HIT, the formation of the heparin–platelet factor 4 (PF4) immune complex leads to the activation of platelets, monocytes, and endothelial cells and the release of tissue factor [68,69].

As a result, blood coagulation is hyperactivated and the risk of both arterial and venous thrombosis increases [70]. Immune complexes (heparin–PF4) can also be detected in APLA-positive patients who have not taken heparin, due to the formation of autoantibodies to PF4.

CAPS is a thrombotic microangiopathic syndrome (TMA) and requires differential diagnosis in various urgent conditions. TMA includes heterogeneous clinical conditions characterized by arteriolar and capillary thrombosis with the development of Coombs-negative hemolytic anemia, thrombocytopenia, and ischemic organ damage resulting from microvascular occlusion by thrombi up to multiple organ failure. Systemic thrombosis occurs as a result of vascular occlusion caused by the hyperaggregation of platelets or the increased formation of fibrin. Hemolysis presents with schizocytosis, which occurs due to the fragmentation of red blood cells by fibrin threads in thrombosed small vessels (capillaries, precapillaries, and terminal arterioles).

CAPS in pregnancy creates many diagnostic challenges due to its wide range of clinical manifestations and similarities with other obstetric complications and microangiopathic diseases, such as hemolytic uremic syndrome (HUS), HELLP syndrome, thrombotic thrombocytopenic purpura (TTP), severe pre-eclampsia, DIC, septic shock, and SIRS [71,72]. Skin necrosis is one of the most striking clinical symptoms of CAPS.

A positive APLA titer in patients with TMA is a poor prognostic sign [73,74].

**Table 3 ijms-25-00668-t003:** Differential diagnosis of CAPS.

Clinical or Laboratory Feature	CAPS	Sepsis [75,76,77]	aHUS [78]	DIC Syndrome [79]
Course of the disease	Up to 1 week	Days	Days	Days
Multiorgan involvement	+	+	+	+
Neurological symptoms	Moderate/severe	Moderate/severe	Minimal/moderate	Moderate/severe
Cardiac involvement	Possible	Possible	Minimal	Possible
Renal involvement	Possible	Possible	Possible	Possible
Fever	±	+	±	±
APLA	+	±	−	−
Hemolytic anemia	±	±	+	±
Schistocytes	±	±	+	±
Thrombocytopenia	Moderate/severe	Moderate/severe	Moderate	Moderate/severe
Fibrinogen	Normal	Normal/reduced	Normal	Reduced
Complement activation	+	+	+	+
Infection as a trigger	±	+	+	+
Treatment	Anticoagulants, intravenous corticosteroids, plasmapheresis, IVIG, resolution of trigger/underlying cause			Anticoagulants, resolution of trigger/underlying cause

## 5. Principles of Treatment

Due to the severity of clinical manifestations and high mortality in CAPS, the choice of adequate therapy is extremely important.

The primary goal of therapy should be to suppress thromboinflammation by reducing excessive complement activation, cytokine levels, NET formation, and excessive thrombin generation. Treatment should not be delayed unless all CAPS criteria are met, especially if there is a history of APS. If CAPS is suspected, aggressive therapy should be initiated immediately. Because CAPS occurs in only 1% of patients with APS, recommendations are based on case reports and expert opinion. An analysis of more than 500 cases of the CAPS registry [35] formed the basis for first-line therapy for CAPS, including (Table 4):(1)Trigger elimination (termination of pregnancy, treatment of infection, etc.) if possible;(2)Full anticoagulation with LMWH at a dose of 1 mg/kg (in the case of enoxaparin) every twelve hours;(3)Glucocorticoids;(4)Plasmapheresis with plasma exchange.

This approach reduced mortality by 37% and increased the recovery rate by 78% [35].

IVIG therapy in addition to triple therapy is included in the international consensus guidelines for CAPS. There are no absolute recommendations regarding the dose or duration of IVIG administration, but it is generally recommended after the last day of plasmapheresis. Two common regimens have been proposed: 2 g/kg body weight for 2–5 days or 400 mg/kg for 5 days [80].

A study that assessed the clinical significance of laboratory findings in patients with CAPS included 14 patients [81]. Triple APLA positivity (IgG/IgM anticardiolipin + IgG/IgM anti-β2-glycoprotein I + lupus anticoagulants) was significantly more prevalent in patients with CAPS compared with controls (*p* = 0.003). All but one of the patients participating in the study received triple therapy, which included various anticoagulants, plasmapheresis, and high doses of corticosteroids. Intravenous immunoglobulins (IVIGs) were included in therapy of six patients. Based on treatment results, all patients with CAPS were discharged after several months.

The severity of the patient’s condition and the results of laboratory and instrumental studies may directly determine the specific therapy. The CAPS treatment with plasmapheresis has not been examined in detail either in controlled clinical trials or in systematic reviews. The estimated number of patients receiving plasmapheresis as part of combination therapy is approximately 100–300 [82].

In a clinical case of CAPS secondary to immune thrombocytopenic purpura (ITP), a 20-year-old patient presented to the emergency department with non-specific manifestation [83]. In this case, treatment with high doses of corticosteroids and the introduction of IVIG did not lead to significant improvements. Several weeks prior to emergency admission, the patient was prescribed adjunctive therapy with the thrombopoietin receptor agonist (TPO-RA) avatrombopag at a dose of 20 mg daily. The examination results indicated respiratory failure. Upon auscultation of the lungs, bilateral wheezing was detected. Additionally, the patient reported experiencing binocular blurred vision. The diagnosis of probable CAPS was based on laboratory results and thrombotic involvement of at least three target organs. The patient also had circulated high-risk antiphospholipid antibodies (LA and aCl IgG positive) at age 9 years. Due to the patient’s serious condition, heparin, vitamin K, IVIG, and high-dose intravenous corticosteroids were prescribed, and avatrombopag therapy was interrupted. The platelet count remained within the reference levels from 200,000/μL to 400.00/μL, and the patient’s well-being significantly improved.

Based on the pathophysiological mechanisms of CAPS development, the use of immunosuppressive drugs is the most reasonable choice [83]. Immunosuppression reduces antibody production, normalizing other B cell functions.

Rituximab is a monoclonal antibody drug that depletes CD20 memory cells and naïve B cells [84]. There are limited data on using rituximab in CAPS and further pharmacokinetic studies are needed. Some features of CAPS may be associated with high levels of acute phase proteins and cytokines, which require a decrease in circulating B cells and administration of rituximab [85].

A report has been made on the effectiveness of an alternative therapy for a patient with CAPS who has a history of HIT [85]. A 43-year-old man was admitted to the intensive care unit with complaints of pain in the lower extremities, a purpuric mesh rash on the right leg, and chest pain. HIT was diagnosed 8 months before presentation. After starting heparin, platelet levels began to fall rapidly, and IgG antibodies to platelet factor 4 were elevated. Laboratory tests revealed positive aCL and anti-β2GPI. Skin biopsy showed vasculopathy with microthrombi, and intensive treatment of CAPS with glucocorticosteroids and rituximab was started. After the second hospitalization due to deterioration of the patient’s condition, the therapy was supplemented with plasmapheresis with plasma as replacement fluid (once daily for 3 consecutive days and a fourth time 3 days after the third plasmapheresis) and argatroban due to previous HIT. While taking large doses of glucocorticosteroids, the patient developed avascular necrosis (AVN) of the right tibia and right foot. Methylprednisolone, sirolimus, and fondaparinux were prescribed. CAPS was suspected in this clinical case due to the appearance of diffuse alveolar hemorrhage, for which the patient was started on mycophenolate, hydroxychloroquine, and methylprednisolone. The patient went into remission and was discharged with recommendations. In cases of severe and persistently relapsing APS, sirolimus may be justified, but data on its use are very limited. However, the literature suggests that sirolimus may be beneficial for certain patient groups and could have a protective effect in patients with APS-associated nephropathy [86].

Eltrombopag may be a trigger for CAPS development. Eltrombopag is a hematopoiesis-stimulating drug used in second-line therapy of ITP. Two case reports describe the treatment of patients with SLE and positive antiphospholipid antibodies who developed CAPS after starting eltrombopag [87]. Patients received solumedrol, plasmapheresis, anticoagulants, and rituximab in the intensive care unit. Further research is required to determine the occurrence of CAPS after taking eltrombopag. However, the authors of the article recommend testing for APLA before prescribing eltrombopag to patients with ITP associated with SLE [88].

As each clinical case is unique, other drugs may be added to the triple therapy for CAPS, including IVIG and immunosuppressive drugs (Table 4). There are new potential therapeutic targets in refractory CAPS. New therapeutic approaches include the use of drugs such as complement inhibitors, TLR inhibitors, anti-CD20 inhibitors, peptides, statins, and maybe others.

For example, a clinical case of treating CAPS in a pregnant woman was described earlier. Eculizumab, a recombinant humanized monoclonal antibody against complement protein C5, was added to the first-line therapy [89]. The patient’s laboratory parameters improved and no side effects were observed after starting eculizumab. The patient gave birth to a healthy baby at 32 weeks of gestation. Inhibition of terminal complement activation is safe and may be effective in patients with CAPS.

**Table 4 ijms-25-00668-t004:** Treatment options in patients with CAPS.

First-Line Treatment [35]	Second-Line Treatment	Experimental Treatment
Heparin; Glucocorticoids; Plasmapheresis/plasma exchange; Intravenous immunoglobulin (IVIG)	Fibrinolytics [90]; Cyclophosphamide [91]; Prostacyclin [90]; Defibrotide [92]; Ancrod [93]	Anticytokine therapy; Direct oral anticoagulants (dabigatran) [94]; Immunosuppressive drugs (mycophenolate, hydroxychloroquine, sirolimus)

## 6. Prognosis and Prevention

### 6.1. Prognosis

Due to the increasing number of new clinical cases of patients with CAPS, the European APLA Forum has established an international registry of patients with CAPS (CAPS Registry), which currently includes approximately 600 patients. High mortality of patients with CAPS is associated with SLE, lung and kidney damage, and also depends on the age of the patients (over 36 years old) and the APLA titer [6]. Other long-term manifestations of CAPS include thrombosis and microantigopathic hemolytic anemia. There are not many cases of relapse of CAPS—18 episodes of “relapse” in eight patients were recorded.

The mortality rate in CAPS has decreased from 50% to approximately 30%, possibly due to improved diagnosis and treatment of CAPS [35,95].

aCL-IgG/IgM, anti-β2GPI, and LA in patients with thrombocytopenia (OR 2.93, 95% CI 1.31–6.56, *p* = 0.009) were associated with increased mortality rate compared with patients who had APS without thrombocytopenia [96]. According to a ten-year retrospective study of 1000 patients with APS, 93 (9.3%) died. The most common causes of death were major thrombosis (36.5%) and infection (26.9%) [97]. CAPS developed in nine patients (0.9%), five of them (55.6%) died.

### 6.2. Prevention

CAPS is a life-threatening condition that, despite treatment, is still associated with high mortality among patients and requires the development of effective preventive measures. These include identifying patients with risk factors: the presence of a high-risk APLA profile, t-APS and/or o-APS with high APLA titers, perinatal or perioperative withdrawal of anticoagulants, and infection. Prevention of CAPS requires anticoagulant prophylaxis before and after surgical interventions, including even trivial manipulations (biopsy, etc.), even in patients with o-APS and low antibody titers, as well as earlier genetic testing for thrombophilia [9]. The use of LMWH in such cases has advantages not only because of its anticoagulant but also anti-inflammatory properties. Prevention and treatment of infection and timely delivery in high-risk groups of pregnant women also seem warranted.

A systematic review of 112 patients analyzed the effect of anticoagulant therapy in patients with APS and the incidence of future CAPS [98]. Inappropriate therapy with vitamin K antagonists, LMWHs, and direct oral anticoagulants may contribute to the development of CAPS in patients with thrombotic APS.

As it is still not fully understood why some patients with APS develop a catastrophic form with multiple organ failure, it is important to look out for the following features:(1)The need to treat any infectious process with antibiotics in these patients is discussed. However, such patients should undergo dynamic monitoring.(2)If a patient with APS requires surgical treatment, oral anticoagulants should be continued as long as possible.(3)In the postpartum period, it is also necessary to receive anticoagulant therapy for 6 weeks.(4)It is necessary to monitor and prevent relapses of SLE [99].

Practitioners should be ready for a catastrophic scenario in all patients with APS in late pregnancy and labor/postpartum due to the significant increase in procoagulant and proinflammatory tendencies during this time. Pregnant women with pre-eclampsia/HELLP syndrome and/or infection are at high risk of developing uncontrolled thromboinflammation and thrombotic storm in CAPS.

### 6.3. CAPS and SARS-CoV-2

A severe SARS-CoV-2 infection is characterized by an hyperinflammatory reaction, endothelial dysfunction and systemic microthrombosis followed by multiple organ failure, increased ferritin, and other indicators, which are associated with high mortality among patients [100]. Clinical manifestation of thrombophilia that occurs during severe SARS-CoV-2 infection is similar to the symptoms of CAPS. SARS-CoV-2 can lead to the development of CAPS in APS-positive patients due to indirect activation of complement and the associated procoagulant state [101]. The mechanisms of hemostasis disorders in severe SARS-CoV-2 infection and CAPS are diverse and similar at the same time. There is often simultaneous activation of many pathways, including activation of platelets, external and internal pathways of coagulation cascade, suppression of fibrinolysis, etc. with the development of DIC, thrombotic microangiopathy, and even local intravascular coagulation in the lungs. Poorly controlled inflammation and hyperactivation of blood clotting cause not only a cytokine storm but also a thrombotic storm in severe COVID-19 and CAPS (Figure 4).

At the same time, ferritin, being a marker of SARS-CoV-2 severity, is also elevated in many patients with CAPS. SARS-CoV-2 can lead to disruption of the cytokine response, but the development of autoimmunity in patients with coronavirus infection is still a matter of debate [102]. An analysis of 45 patients with a confirmed diagnosis of coronavirus infections showed that the APLA titer was increased only in a small group of patients (20%), while in the majority of patients (80%) it did not reach the lower limit or was normal [103]. The negative APLA titer did not correlate with the further development of large vessel thrombosis and pulmonary embolism in 11% of patients.

The relationship between SARS-CoV-2 and CAPS is currently difficult to assess due to the limited number of studies [104].

Reports present isolated clinical cases of patients in whom SARS-CoV-2 played a key role in the development of APS [105].

Further research is needed to determine a more precise relationship between severe coronavirus infections and CAPS. There is still no general agreement if the high titer of APLA, when transiently diagnosed with SARS-CoV-2, is associated with thrombotic phenomena in the catastrophic form of antiphospholipid syndrome.

## 7. Conclusions

CAPS is a multifactorial disease that requires an interdisciplinary approach and highly qualified medical care, adequate and timely diagnosis, and appropriate prevention. The variety of clinical symptoms often leads to late diagnosis of this condition with high mortality. Further study of the pathological mechanisms of CAPS and the role of thromboinflammation can significantly expand the prevention and treatment of this life-threatening condition through the use of capable drugs that can interrupt the crucial pathological pathways.

## 8. Take-Home Message

CAPS is a rare life-threatening disease that requires urgent aggressive therapy even if not all criteria of CAPS are fulfilled;Main risk factors for developing the CAPS are infection, pregnancy and postpartum period, surgery or trauma, anticoagulant withdrawal, oral contraception, genetic thrombopilia, vaccination, ART;CAPS is a thrombotic microangiopathic syndrome (TMA) and can mask many life-threatening conditions such as HELLP syndrome, severe pre-eclampsia, sepsis and septic shock, and other TMAs;Severe SARS-CoV-2 infection and CAPS have similarities in pathogenesis and clinical manifestations: thromboinflammation, immunothrombosis, thrombotic storm, and multiple organ failure;First-line therapy of CAPS includes: trigger elimination, anticoagulants, glucocorticoids, and plasma exchange.

## Figures and Tables

**Figure 1 ijms-25-00668-f001:**
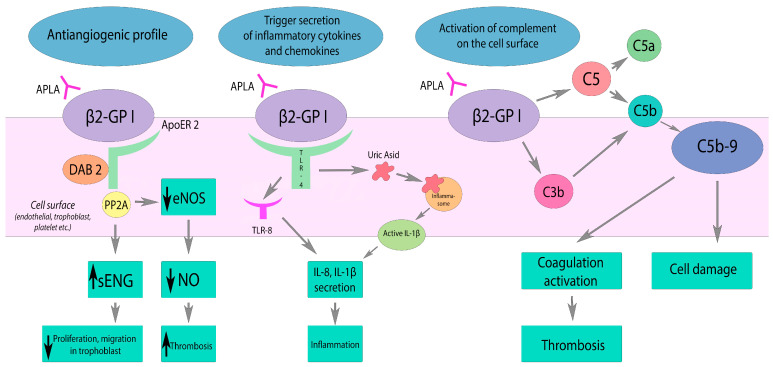
Effects of APLA on the complement system, inflammation, and vascular tone. β2GP1—β2-glycoprotein I, ApoER2—apolipoprotein E2 receptor, DAB2—Disabled-2, PP2A—Protein phosphatase 2A, eNOS—nitric oxide synthase, sENG—soluble Endoglin, TLR-4—toll-like receptor 4, TLT-8—toll-like receptor 8. APLA reduces eNOS activity via ApoER2 interaction. Decreased NO production results in impaired vasodilatation and endothelial dysfunction. APLA activates TLR and inflammasome pathways, triggering the secretion of inflammatory cytokines and chemokines. APLA activates complement on the cell surface, leading to the coagulation activation and cell damage by C5b-9 deposition. Black arrow down—decrease; black arrow up—increase.

**Figure 2 ijms-25-00668-f002:**
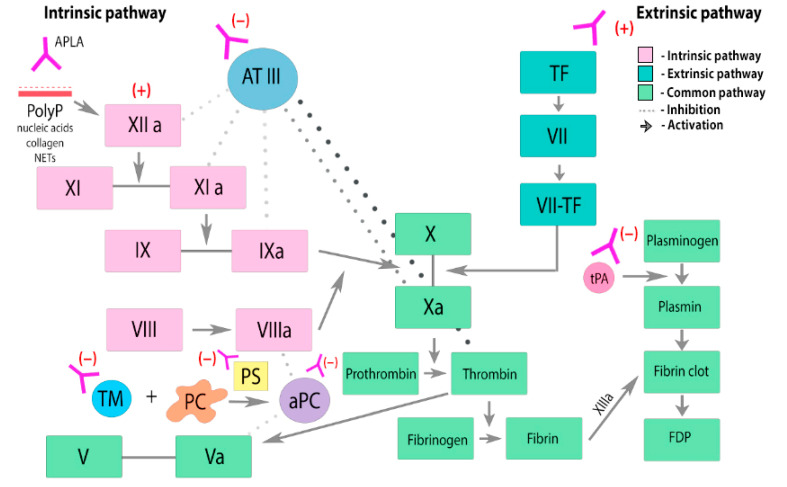
Effects of APLA on the coagulation cascade and fibrinolysis. PC—protein C. PS—protein S. aPC—activated protein C. TF—tissue factor. ATIII—antithrombin III. PolyP—anionic polyphosphates. TM—thrombomodulin. APLA prevents thrombin inactivation by inhibiting antithrombin III. APLA may interfere with the components of protein C activation pathway and inhibit its anticoagulant effect, leading to an increased risk of thrombosis. APLA inhibits tPA-mediated conversion of plasminogen to plasmin, leading to dysregulation and suppression of fibrinolysis.

**Figure 3 ijms-25-00668-f003:**
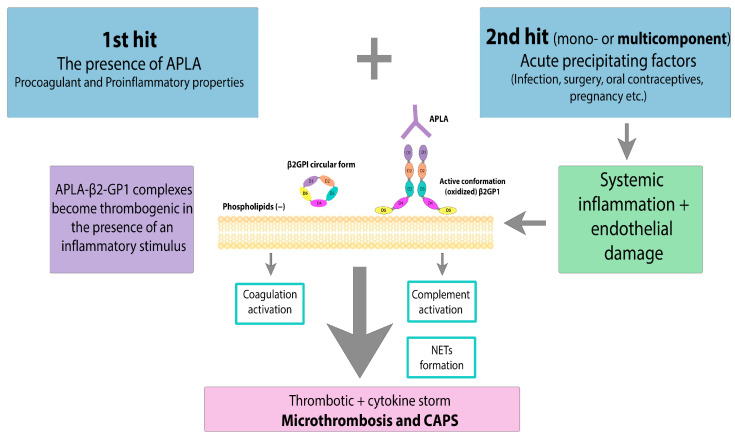
CAPS: the “two-hit” theory. APLA—antiphospholipid antibody, β2GPI—β2-glycoprotein I. Acute precipitating factors as a 2nd hit lead to systemic inflammation and endothelial damage. Massive cytokine release, NET formation and coagulation activation result in microthrombosis and CAPS.

**Figure 4 ijms-25-00668-f004:**
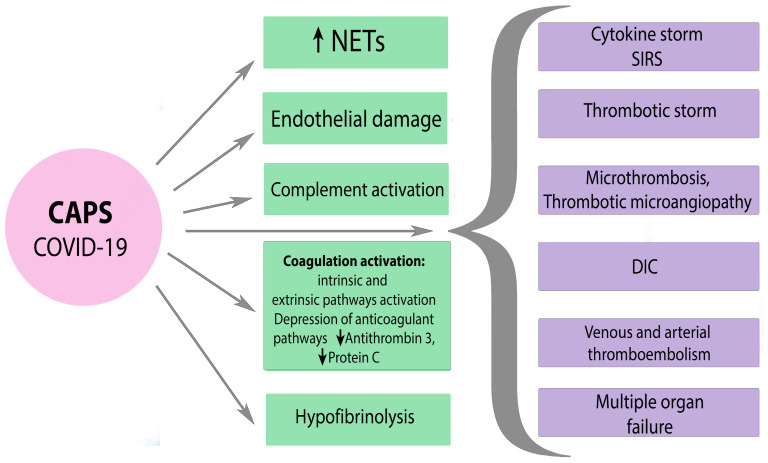
The mechanisms of hemostasis disorders in severe SARS-CoV-2 and CAPS. CAPS—catastrophic APS, NETs—neutrophil extracellular traps, SIRS—systemic inflammatory response syndrome, DIC—disseminated intravascular coagulation. Black arrow down—decrease; black arrow up—increase.

**Table 1 ijms-25-00668-t001:** CAPS classification criteria [4].

Clinical manifestation of damage to 3 or more organs and systems; Simultaneous and rapid development of clinical manifestations (less than 1 week); Occlusion of small-caliber vessels on histological examination; Laboratory confirmation of the presence of APLA (according to the Sydney recommendations);
**The diagnosis of a specific CAPS should meet all 4 diagnostic criteria ***
*A diagnosis of probable CAPS ** is made if:* — *Criteria 2, 3, and 4 are present, but only two organs or systems are involved in the pathological process, instead of 3 and more;* – *Criteria 1, 2, and 3 are observed, but there is no laboratory confirmation of the presence of APLA;* – *Criteria 1, 2, and 4 are present;* – *Criteria 1, 3, and 4 are present, but clinical manifestations develop within 1 month, despite anticoagulant therapy.*

* A diagnosis of definite CAPS is made when a patient meets all 4 criteria. ** Potential CAPS is diagnosed when one of the criteria is absent/incomplete.

**Table 2 ijms-25-00668-t002:** The variability of clinical manifestation and outcomes in CAPS.

Author	Number of Patients	Clinical Features	Outcome
Rodríguez-Pintó I et al. [35]	500	Skin, lungs, kidneys, central nervous system, heart, and gastrointestinal organs are the most often affected	Mortality accounted for 37% of episodes
Dupré et al. [36]	120	65 patients with cutaneous involvement, others with at least 1 episode of non-cutaneous CAPS	Mortality did not differ between the groups (respectively, 5% vs. 9%, *p* = 0.47)
Sanchez et al. [37]	1	17-year-old girl presented with fatigue, loss of appetite, arthralgia, lower limb skin ulcers, and livedo reticularis	Within 2 months of treatment CAPS resolved
Mardani et al. [38]	1	Sudden renal failure	Chronic renal insufficiency required dialysis
Morel N et al. [40]	11	Ophthalmic manifestations in CAPS were usually bilateral (*n* = 19, 83%), most often occlusive retinopathy (*n* = 17, 74%), choroidopathy (*n* = 11, 48%), or retinal vasculitis (*n* = 1, 4%)	Nearly half the patients had permanent vision loss
Dorji et al. [31]	3	Case 1—SLE presented with anasarca, abnormal mentation, pulmonary thromboembolism involving right middle and left lower lobes. Patient developed infarct in left middle cerebral artery territory Case 2—oral ulcers, alopecia, photosensitive malar rash, polyarthritis, Raynaud’s phenomenon, intermittent fever with headache, and arterial thrombosis resulting in gangrene of the right thumb Case 3—thrombosis infra-renal abdominal aorta and in the left popliteal artery	Mortality rate 66.6%
Carvalho et al. [44]	1	Acute peritonitis due to perforation of the sigmoid	SLE diagnosed after the CAPS episode
Nawata et al. [45]	1	Renal, intestinal, and pulmonary infarction	Mortality rate—100%
Christiansen et al. [46]	1	Abdominal pain, vomiting, pneumoperitoneum, small intestinal necrosis	Resolved after 72 days of treatment
McRae et al. [47]	1	Pulmonary embolism, subclavian vein thrombosis, epidermal and pandermal necrosis of the right hand	CAPS episode resolved
Girish B et al. [48]	1	Grade 1 hypertensive retinopathy, mitral regurgitation, pre-eclampsia with partial HELLP syndrome, cardiac arrest	Mortality rate—100%
Gómez-Puerta JA et al. [54]	15	CAPS occurred during pregnancy in 7 patients. In 8 (53%), HELLP syndrome was established, and in 4 (27%) placental infarction. Pelvic vein thrombosis and myometrial TMA in 1 patient.	Mortality rates were high among both mothers (46%) and infants (54%)
Bulvik S et al. [55]	1	Gestational hypertension in 33rd week of gestation. Within 3 days after C-section, the patient developed pallor, purpura, wound dehiscence	Asymptomatic after 1 episode of CAPS
Knickerbocker WJ et al. [56]	1	Patient two days postpartum. She complained of multiple aches and pains, worst in the abdomen and lower back. A bone biopsy showed extensive marrow necrosis	Clinically recovered after the treatment. Has radiologic and histologic evidence of a progressive myelofibrosis and myelosclerosis
Sinha J et al. [57]	1	Fatal bone marrow necrosis along with involvement of liver, lung, and central nervous system during previous pregnancy	Mortality rate—100%

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
