# Peer review of "Catastrophic Antiphospholipid Syndrome"

_ijms, 2024, doi:10.3390/ijms25010668_

Round 1
Reviewer 1 Report
Comments and Suggestions for Authors
Authors reviewed Catastrophic antiphospholipid syndrome.
This manuscript is potentially interesting, several issues arise.
There are many abbreviations. Authors should explain those.
There are no take home message or new findings
“Prognosis” and “prevention” section are not sufficient.
Figure or table for “treatment” may be helpful.
Efficacy of treatment may be helpful.
Table 1. CAPS classification criteria needs improvements.
1) Reference should be required.
2) Explanation should be required in legend.
3) “all criteria are observed, but only two organs or systems are involved in the pathological process” is not clear.
4) “all four criteria are observed, but there is no laboratory confirmation of the circulation of APLA” is not clear.
Table 2 should be remade and needs reference.
Table 3 should be remade and needs explanation.
Author Response
Thank you very much for taking the time to review this manuscript. Please read the detailed responses below and the relevant changes/corrections. Comments 1: There are many abbreviations. Authors should explain those. Response 1: All abbreviations have been reviewed and corrected. We deciphered all the abbreviations. Comments 2: There are no take home message or new findings. Response 2: The take-home message was added at the end of the article. Comments 3: “Prognosis” and “prevention” sections are not sufficient. Response 3: This section has been expanded with additional information. Comments 4: Figure or table for “treatment” may be helpful. Response 4: New information, the additional table and the figure, added to the treatment section. Comments 5: Table 1. CAPS classification criteria needs improvements. Response 5: Transcripts and additional information were added to table 1. The ambiguous sentences have been corrected. Comments 6: Table 2 should be remade and needs reference. Response 6: We decided to revise Table 2 and presented it as a figure. Also, we attached a link to the research. Comments 7: Table 3 should be remade and needs explanation. Response 7: Table 3 has been updated with additional sources and new information.
Reviewer 2 Report
Comments and Suggestions for Authors
The manuscript is a narrative review regarding the pathology, diagnosis and management of the catastrophic antiphospholipidic syndrome. While the subject is not necessarily new, and it was recently discussed in the context of Covid-19 pandemic, the review brings together the current knowledge on this subject.
The authors present anarrative review about the pathogeny anf treatment of the catastrofic antiphospholipic syndrome. The article is generally well structured and up-to date. Earlu recognition of this pathology may be life-saving.
Some minor errors should be corected.
In the Introduction, the authors should add more information about the methodology of selection of the articles included in the review (databases, key words used, period of time).
A figure to better illustrate the pathological pathways involved in CAS should be added in section 2
In section 3, a table with the studied included in the review and cited should be added, with the number of patients, clinical features, outcomes.
Also, appropiate reference should be added after each name cited (Dupret, Sanchez, etc..)
The references are appropriate, and most of them are new (within the last 5 years).
The conclusions reflect the major findings in the article.
Minor English corrections are required.
Author Response
Thank you very much for taking the time to review this manuscript. Please read the detailed responses below and the relevant changes/corrections. Comments 1: In the Introduction, the authors should add more information about the methodology of selection of the articles included in the review (databases, keywords used, period of time). Response 1: According to your comment, we have added methodology data in the introduction section. Comments 2: A figure to better illustrate the pathological pathways involved in CAS should be added in section 2. Response 2: We have added pictures In Section 2, that represent key aspects in the pathogenesis of CAPS. Comments 3: In section 3, a table with the studied included in the review and cited should be added, with the number of patients, clinical features, outcomes. Response 3: We have added a table with the listed information in section 3. Comments 4: Also, appropriate reference should be added after each name cited (Dupret, Sanchez, etc..) Response 4: References for each quoted name are added.Round 2
Reviewer 1 Report
Comments and Suggestions for Authors
Revised manuscript needs minor revision.
Figure 1 is not informative. APS may include other conditions.
Figure 3 may be helpful to include PC, PS, AT and TM.
Reformatting of Table 2 may be helpful.
Figure 5 is not new. It is better to be deleted.
Figure 6 is not new. It is better to be deleted or changed to Table 4..
Author Response
Dear reviewer, thank you for your valuable suggestions on the article! Comments 1. Figure 1 is not informative. APS may include other conditions. Response 1: We have removed Figures 1. Comments 2. Figure 3 may be helpful to include PC, PS, AT and TM. Response 2. We have included PC, PS, and TM in Figure 3 with explanations. Comments 3. Reformatting of Table 2 may be helpful. Response 3. Table 2 has been reformatted. Comments 4. Figure 5 is not new. It is better to be deleted. Figure 6 is not new. It is better to be deleted or changed to Table 4. Response 4. We have removed Figures 5 and 6. We have added information about first-line therapy in Table 4.